# SBEVNet: End-to-End Deep Stereo Layout Estimation

## Abstract

Accurate layout estimation is crucial for planning and navigation in robotics applications, such as self-driving. In this paper, we introduce the Stereo Bird's Eye View Network (*SBEVNet*), a novel supervised end-to-end framework for estimation of bird's eye view layout from a pair of stereo images. Although our network reuses some of the building blocks from the state-of-the-art deep learning networks for disparity estimation, we show that explicit depth estimation is neither sufficient nor necessary. Instead, the learning of a good internal bird's eye view feature representation is effective for layout estimation. Specifically, we first generate a disparity feature volume using the features of the stereo images and then project it to the bird's eye view coordinates. This gives us coarse-grained information about the scene structure. We also apply inverse perspective mapping (IPM) to map the input images and their features to the bird's eye view. This gives us fine-grained texture information. Concatenating IPM features with the projected feature volume creates a rich bird's eye view representation which is useful for spatial reasoning. We use this representation to estimate the BEV semantic map. Additionally, we show that using the IPM features as a supervisory signal for stereo features can give an improvement in performance. We demonstrate our approach on two datasets: the KITTI (Geiger et al., 2013) dataset and a synthetically generated dataset from the CARLA (Dosovitskiy et al., 2017) simulator. For both of these datasets, we establish state-of-the-art performance compared to baseline techniques. [1]

## 1 Introduction

Layout estimation is an extremely important task for navigation and planning in numerous robotics applications such as autonomous driving cars. The bird's eye view (BEV) layout is a semantic occupancy map containing per pixel class information, e.g. road, sidewalk, cars, vegetation, etc. The BEV semantic map is important for planning the path of the robot in order to prevent it from hitting objects and going to impassable locations.

In order to generate a BEV layout, we need 3D information about the scene. Sensors such as LiDAR (Light Detection And Ranging) can provide accurate point clouds. The biggest limitations of LiDAR are high cost, sparse resolution, and low scan-rates. Also, as an active sensor LiDAR is more power hungry, more susceptible to interference from other radiation sources, and can affect the scene. Cameras on the other hand, are much cheaper, passive, and capture much more information at a higher frame-rate. However, it is both hard and computationally expensive to get accurate depth and point clouds from cameras.

The classic approach for stereo layout estimation contains two steps. The first step is to generate a BEV feature map by an orthographic projection of the point cloud generated using stereo images. The second step is bird's eye view semantic segmentation using the projected point cloud from the first step. This approach is limited by the estimated point cloud accuracy because the error in it will propagate to the layout estimation step. In this paper, we show that explicit depth estimation is actually neither sufficient nor necessary for good layout estimation. Estimating accurate depth is not sufficient because many areas in the 3D space can be occluded partially, e.g. behind a tree trunk.

---

[1]The code and the synthesized dataset will be made public upon the acceptance of this paper.

However, these areas can be estimated by combining spatial reasoning and geometric knowledge in bird's eye view representation. Explicitly estimating accurate depth is also not necessary because layout estimation can be done without estimating the point cloud. Point cloud coordinate accuracy is limited by the 3D to 2D BEV projection and rasterization. For these reasons, having an effective bird's eye view representation is very important.

*SBEVNet* is built upon recent deep stereo matching paradigm. These deep learning based methods have shown tremendous success in stereo disparity/depth estimation. Most of these models (Liang et al., 2018; Khamis et al., 2018; Wang et al., 2019b; Sun et al., 2018; Guo et al., 2019; Zhang et al., 2019; Chang & Chen, 2018; Kendall et al., 2017) generate a 3-dimensional disparity feature volume by concatenating the left and right images shifted at different disparities, which is used to make a cost volume containing stereo matching costs for each disparity value. Given a location in the image and the disparity, we can get the position of the corresponding 3D point in the world space. Hence, every point in the feature volume and cost volume corresponds to a 3D location in the world space. The innovation in our approach comes from the observation: it is possible to directly use the feature volume for layout estimation, rather than a two step process, which uses the point cloud generated by the network. We propose *SBEVNet*, an end-to-end neural architecture that takes a pair of stereo images and outputs the bird's eye view scene layout. We first project the disparity feature volume to the BEV view, creating a 2D representation from the 3D volume. We then warp it by mapping different disparities and the image coordinates to the bird's eye view space. In order to overcome the loss of fine grained information imposed by our choice of the stereo BEV feature map, we concatenate a projection of the original images and deep features to this feature map. We generate these projected features by applying inverse perspective mapping (IPM) (Mallot et al., 1991) to the input image and its features, choosing the ground as the target plane We feed this representation to a U-Net in order to estimate the BEV semantic map of the scene.

In order to perform inverse perspective mapping, we require information about the ground in the 3D world space. Hence we also consider the scenario where we perform IPM during the training time and not the inference time. Here, during the training time, we use cross modal distillation to transfer knowledge from IPM features to the stereo features.

*SBEVNet* is the first approach to use an end-to-end neural architecture for stereo layout estimation. We show that *SBEVNet* achieves better performance than existing approaches. *SBEVNet* outperforms all the baseline algorithms on KITTI (Geiger et al., 2013) dataset and a synthetically generated dataset extracted from the CARLA simulator (Dosovitskiy et al., 2017). In summary, our contributions are the following:

1. We propose *SBEVNet*, an end-to-end neural architecture for layout estimation from a stereo pair of images.

2. We learn a novel representation for BEV layout estimation by fusing projected stereo feature volume and fine grained inverse perspective mapping features.

3. We evaluate *SBEVNet* and demonstrate state-of-the-art performance over other methods by a large margin on two datasets – KITTI dataset and our synthetically generated dataset using the CARLA simulator.

## 2 RELATED WORK

To the best of our knowledge, there is no published research paper for estimating layout given a pair of stereo images. However, there are several works tackling layout estimation using a single image or doing object detection using stereo images. In this section, we review the most closely related approaches.

**Monocular Layout Estimation** MonoLayout (Mani et al., 2020) uses an encoder-decoder model to estimate the bird's eye view layout using a monocular input image. They also leverage adversarial training to produce sharper estimates. MonoOccupancy (Lu et al., 2019) uses a variational encoder-decoder network to estimate the layout. Both MonoLayout and MonoOccupancy do not use any camera geometry priors to perform the task. Schulter et al. (2018) uses depth estimation to project the image semantics to bird's eye view. They also use Open Street Maps data to refine the BEV images via adversarial learning. Wang et al. (2019c) uses (Schulter et al., 2018) to estimate the parameters of the road such as lanes, sidewalks, etc. Monocular methods learn strong prior, which

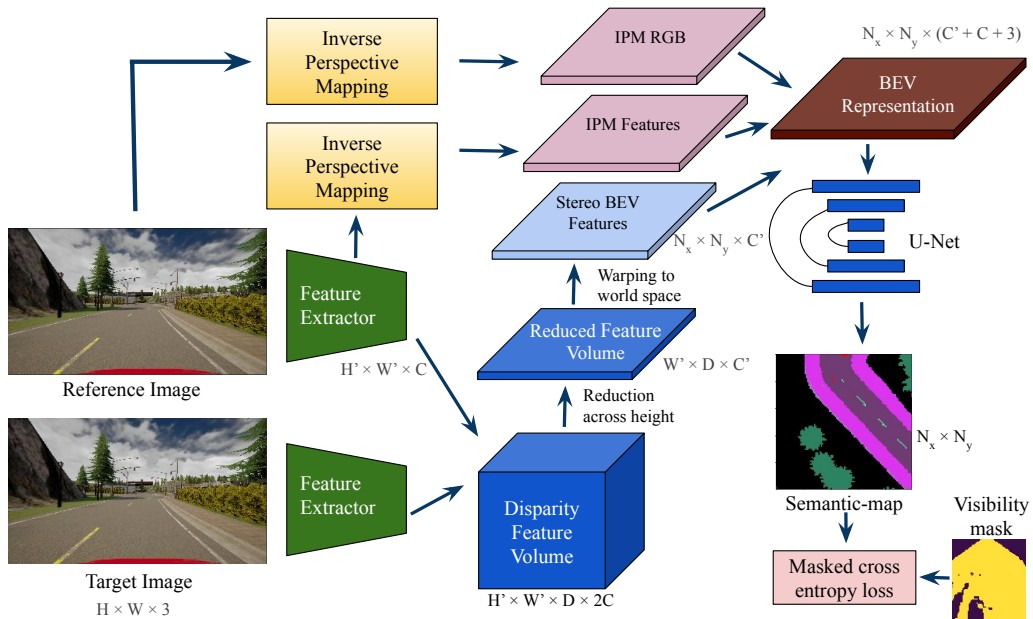

Figure 1: *SBEVNet* overview. We first extract the image features given the target and reference image. Using the pair of features, we create a disparity feature volume. We then reduce the disparity feature volume along with the height and warp it in the bird's eye view layout space. On a parallel branch of the network, we apply inverse perspective mapping (IPM) on the reference image and its features. We concatenate the IPM RGB, IPM feature, and the stereo BEV features. The BEV representation is then used to estimate the semantic map through a U-Net. Visibility mask is used to apply the supervised loss only at the locations in the BEV which are in the view of the front camera.

does not generalize well when there is a significant domain shift. Stereo methods learn weak-prior plus geometric relationship, which can generalize better.

**Deep Stereo Matching** Several methods like (Liang et al., 2018; Khamis et al., 2018; Wang et al., 2019b; Sun et al., 2018; Guo et al., 2019; Zhang et al., 2019; Chang & Chen, 2018; Kendall et al., 2017) extract the features of the stereo images and generate a 3D disparity feature volume for disparity/depth estimation. They use a 3D CNN on the feature volume to get cost volume to perform stereo matching. PSMNet (Chang & Chen, 2018) uses a spatial pyramid pooling module and a stacked hourglass network to further improve the performance. High-res-stereo (Yang et al., 2019) uses a hierarchical model, creating cost volumes at multiple resolutions, performing the matching incrementally from over a coarse to fine hierarchy.

**Bird's Eye View Object Detection** Several approaches like Yang et al. (2018); Shi et al. (2019) use LiDAR to perform 3D object detection. Pseudo-lidar (Wang et al., 2019a) and pseudo-LiDAR++ (You et al., 2019) use stereo input to first generate a 3D point cloud and then use a 3D object detection network (Ku et al., 2018; Yang et al., 2018; Shi et al., 2019) on top. BirdGAN (Srivastava et al., 2019) maps the input image to bird's eye view using adversarial learning. The closest work to our approach is DSGN (Chen et al., 2020) which constructs a depth feature volume and map it to the 3D space which is then projected to bird's eye view to perform object detection. The task of object detection is of sparse prediction, whereas layout estimation is of dense fine granularity prediction. Hence we introduced IPM to fuse low level detail with the stereo information to improve the performance of layout estimation.

## 3  OUR METHOD

This section describes the detailed architecture of our proposed framework. *SBEVNet* is built upon recent deep stereo matching paradigms and follows the rules of multi-view camera geometry. An overview of the *SBEVNet* is summarized in Figure 1.

## 3.1 PROBLEM FORMULATION

In this paper, we address the problem of layout estimation from a pair of stereo images. Formally, given a reference camera image $I_R$ and a target camera image $I_T$ both of size $H \times W \times 3$, the camera intrinsics $K$, and the baseline length $T_b$, we aim to estimate the bird's eye view layout of the scene. In particular, we estimate the BEV semantic map of size $N_x \times N_y \times N_C$ within the rectangular range of interest area $(x_{min}, x_{max}, y_{min}, y_{max})$ in front of the camera. Here $H$ is image height, $W$ is image width, and 3 indicates RGB channels. $N_x$ and $N_y$ are the number of horizontal cells and vertical cells respectively in bird's eye view. $N_C$ is the number of semantic classes. This BEV semantic map contains the probability distribution among all semantic classes at each cell of the layout. We assume that the input images are rectified.

## 3.2 FEATURE EXTRACTION

The first step for *SBEVNet* is to extract features $F_R$ and $F_T$ of size $H' \times W' \times C$ for the reference image and the target image respectively. This is done by passing $I_R$ and $I_T$ through a convolutional encoder with shared weights. This produces multi-channel down-sized feature representations which are next used for building disparity feature volumes.

## 3.3 DISPARITY FEATURE VOLUME GENERATION

Similar to (Liang et al., 2018; Khamis et al., 2018; Wang et al., 2019b; Sun et al., 2018; Guo et al., 2019; Zhang et al., 2019; Chang & Chen, 2018; Kendall et al., 2017) we form a disparity feature volume $V$ by concatenating the features $F_R$ and $F_T^d$, where $F_T^d$ is $F_T$ shifted horizontally by a disparity of $d$ pixel, resulting in a 3D volume of size $H' \times W' \times D \times 2C$. We then pass the feature volume through a series of 3D convolution layers with skip connections to learn higher level features. This feature volume at each $d \in \{0, 1, \cdots, D-1\}$ contains a representation of the 3D world at the depth corresponding to the disparity $d$. Rather than using this feature volume to do disparity estimation, we project and warp it to form a bird's eye view representation in the next step.

## 3.4 BIRD'S EYE VIEW REPRESENTATION

The bird's eye view representation is composed of two parts – 1) The stereo BEV representation which is derived from the disparity feature volume, 2) The IPM BEV representation which is the result of applying inverse perspective mapping on the reference image and the features of the reference image. These two parts are concatenated to form the final bird's eye view representation.

### 3.4.1 STEREO BEV REPRESENTATION

The disparity feature volume generated is widely used to estimate depth/disparity in the stereo image pairs. But this feature volume contains a lot of information about the 3D scene which can be used for other tasks as well. Each point in the disparity feature volume corresponds to a point in the 3D world space. We first need to map the 3D feature volume to a 2D feature map containing information of the bird's eye view. If we do max/average pooling along height dimension, a certain degree of the height information is lost quickly before being extracted for our task, which is not desirable. Considering height information a good prior for layout estimation but we don't need to recover it explicitly, we concatenate the feature volume along the height, creating a 2D image of size $W' \times D \times 2CH'$. We then use 2D convolutions to generate the reduced feature volume of size $W'' \times D'' \times C'$. This reduced feature volume does not spatially match with the bird's eye view layout. Hence, we warp the reduced feature volume, transforming it to a feature map of size $N_x \times N_y \times C'$ in the bird's eye view space. Given the disparity $d$, position in the image along width $u$, camera parameters $f$, $c_x$, $c_y$, and stereo baseline length $T_x$, we can find the coordinates in the bird's eye space $x'$ and $y'$ as follows:

$$x' = \frac{(u - c_x) \cdot T_x}{d} \tag{1}$$

$$y' = \frac{f \cdot T_x}{d} \tag{2}$$

The 2D origin of the bird's eye view is co-located with the reference camera. An example visualization of the layout in the disparity volume space is shown in Figure 3. After mapping the coordinates to the BEV space, we map them to a grid of size $N_x \times N_y$ giving us the stereo BEV representation $R_{\text{stereo}}$.

### 3.4.2 IPM BEV REPRESENTATION

The stereo BEV representation contains structural information for the bird's eye view space. Due to the refinement and reduction of the feature volume, the fine grained details are excluded by design. To circumvent that, we need to fuse the low level features to the stereo BEV features, while maintaining geometric consistency.

In order to fuse the image features to the stereo BEV features at the correct locations, we need to warp the image features to the BEV space. We apply inverse perspective mapping on the reference image and the features of the reference image to do that.

A point in the image $I_R$ can correspond to multiple points in the 3D world space due to perspective projection, but there is a single point which also intersects with the ground plane. Let $z = ax + by + c$ be the equation of the ground plane in the world space. Given the input image coordinates $(u, v)$ and camera parameters $f, c_x, c_y$, we can find the coordinates in the bird's eye space $x'$ and $y'$ as follows:

$$x' = \frac{cu - cc_x}{ac_x - au - bf - c_y + v} \tag{3}$$

$$y' = \frac{cf}{ac_x - au - bf - c_y + v} \tag{4}$$

This can be easily derived by combining the camera projection equation with the equation of the ground plane. For many applications, the ground is either planar or can be approximated by a plane. This is also equivalent to computing a homography $H$ between the ground plane and the image plane of the layout and then applying the transformation. We can have the parameters of the plane $a$, $b$, and $c$ pre-determined if the placement of the camera with respect to the ground is known, which is the case for many robotics applications. We can also determine $a$, $b$ and $c$ by using stereo depth and a semantic segmentation network for the road/ground class.

Examples of IPM on the input images is shown in Figure 3 in the appendix. We apply the inverse perspective transform on both the input image and the features of the input image to transform them to the bird's eye view space:

$$R_{\text{IPM\_feat}} = \text{IPM}(F_R) \tag{5}$$

$$R_{\text{IPM\_img}} = \text{IPM}(I_R) \tag{6}$$

They are then concatenated with the stereo BEV representation to form the combined BEV representation:

$$R_{\text{BEV}} = [R_{\text{IPM\_feat}}; R_{\text{IPM\_img}}; R_{\text{stereo}}] \tag{7}$$

### 3.4.3 IPM FOR CROSS MODAL DISTILLATION

There can be use-cases where we cannot do inverse perspective mapping during inference time, due to the unavailability of the ground information. Hence, we consider the case where IPM is only available during the training time. We can think of the IPM features and the stereo features as different modalities and apply cross modal distillation (Gupta et al. (2016)) across them, and transfer knowledge from IPM features to the stereo features. Hence, we use the IPM BEV representation as a supervisory signal for the stereo BEV features. This forces the stereo branch of the model to implicitly learn the fine grained information learned by the IPM features. Rather than concatenating the IPM BEV features with the stereo features, we minimize the distance between them. We call this variant of *SBEVNet* as **SBEVNet-CMD** (*SBEVNet* cross modal distillation). During the training time, the IPM BEV features and the stereo features are used to generate the BEV semantic maps.

$$C^{\text{IPM}} = \text{U-Net}(R_{\text{IPM\_feat}}) \tag{8}$$

$$C^{\text{stereo}} = \text{U-Net}(R_{\text{stereo}}) \tag{9}$$

This ensures both IPM BEV features and stereo BEV features learn meaningful information. We jointly minimise the $L_1$ distance between first $K$ channels of the features.

$$L_{\text{KT}} = \|R_{\text{IPM\_feat}}[: K] - R_{\text{stereo}}[: K]\|_{L_1} \tag{10}$$

By this, we ensure that the stereo model can learn information that is not in the IPM features. In our experiments, we found this to yield better results compared to the approach of minimizing the $L_1$ distance between all the channels of the features. During test time, we only use the stereo features to get the BEV layout. Our experiments show that the stereo model with cross modal distillation performs better than the stereo model without cross modal distillation.

### 3.4.4 LAYOUT GENERATION

We can generate the semantic map by inputting the BEV features to a semantic segmentation network. We pass the concatenated stereo BEV feature map and IPM BEV feature map to a U-Net (Ronneberger et al., 2015) network to generate the semantic map $C$.

$$C = \text{U-Net}(R_{\text{BEV}}) \tag{11}$$

Some areas in the layout may not be in the view of the front camera, e.g. things behind a wall. That is why it is not a good idea to penalize the model for the wrong prediction for those areas. Hence, we use a visibility mask to mask the pixel-wise loss, applying it only on the pixels which are in the field of view. This mask is generated during the ground truth generation process by using ray-tracing on the point cloud to determine which are in the field of view. For a visibility mask $V$, $V_i$ is 1 if the pixel $i$ is in the view of the input image, and 0 otherwise. For the loss, we use a pixel-wise categorical cross entropy loss as follows:

$$L_r = \sum_{i \in P} V_i \cdot \text{CCE}(C_i, C_i^h) \tag{12}$$

where $C_i^h$ is ground truth. The total loss for *SBEVNet-CMD* is the sum of supervision loss from the two feature maps and the $L_1$ distance minimization.

$$L_c = \sum_{i \in P} V_i \cdot \text{CCE}(C_i^{\text{IPM}}, C_i^h) + \sum_{i \in P} V_i \cdot \text{CCE}(C_i^{\text{stereo}}, C_i^h) + L_{\text{KT}} \tag{13}$$

## 4 EXPERIMENTS

### 4.1 DATASETS

**CARLA dataset:** We use the CARLA (Dosovitskiy et al., 2017) simulator to generate a synthetic dataset, containing 4,000 and 925 training and testing data points respectively. The bounds of the layout with respect to the camera are -19 to 19 meters in $x$ direction and 1 to 39 meters in the $y$ direction.

**KITTI dataset:** We also evaluate *SBEVNet* on the odometery subset of the KITTI (Geiger et al., 2013) dataset. We use the SemanticKITTI Behley et al. (2019) dataset for labeled ground truth.

More details about the datasets can be found in the Appendix.

### 4.2 EVALUATION METRICS

As not all the regions of the ground truth layout are visible from the camera, we only consider pixels of the layout which are in the field of view. For evaluating the semantic map, we use macro averaged intersection over union (IoU) scores for the layout pixels which are in the visibility mask. We report the IoU scores for each semantic class separately.

### 4.3 COMPARED METHODS

There are no previously reported quantitative results for the task of stereo layout estimation in our setting. Thus, we evaluate appropriate baselines which are prior works extended to our task.

| Method | mIoU | Road | Vegetation | Cars | Sidewalk | Building |
|---|---|---|---|---|---|---|
| Pseudo-LiDAR + segmentation | 25.63 | 37.64 | 16.40 | 35.15 | 25.44 | 13.50 |
| Pseudo-LiDAR + BEV U-Net | 36.61 | 63.55 | 31.87 | 45.64 | 29.97 | 12.01 |
| IPM + BEV U-Net | 32.30 | 66.36 | 15.24 | 41.37 | 32.77 | 5.78 |
| MonoLayout | 22.16 | 52.88 | 9.00 | 16.36 | 23.17 | 9.41 |
| MonoLayout + depth | 21.85 | 52.94 | 9.95 | 14.07 | 23.02 | 9.31 |
| MonoOccupancy + depth | 29.49 | 67.96 | 16.56 | 7.66 | 36.35 | 18.91 |
| *SBEVNet* only stereo | 36.10 | 64.74 | 31.85 | 39.76 | 30.01 | 14.14 |
| *SBEVNet* stereo + RGB IPM | 39.77 | 65.01 | **33.20** | 47.88 | 33.24 | 19.53 |
| *SBEVNet* stereo + features IPM | 42.29 | 71.29 | 29.79 | 51.97 | 38.46 | 19.95 |
| *SBEVNet*-CMD | 40.10 | 69.07 | 32.71 | 45.45 | 35.16 | 18.08 |
| ***SBEVNet*** | **44.36** | **72.82** | 32.07 | **55.32** | **40.69** | **20.78** |
| ***SBEVNet* Ensemble** | **47.92** | **75.36** | **35.33** | **60.17** | **44.25** | **24.47** |

Table 1: Quantitative results of semantic layout estimation on the CARLA dataset.

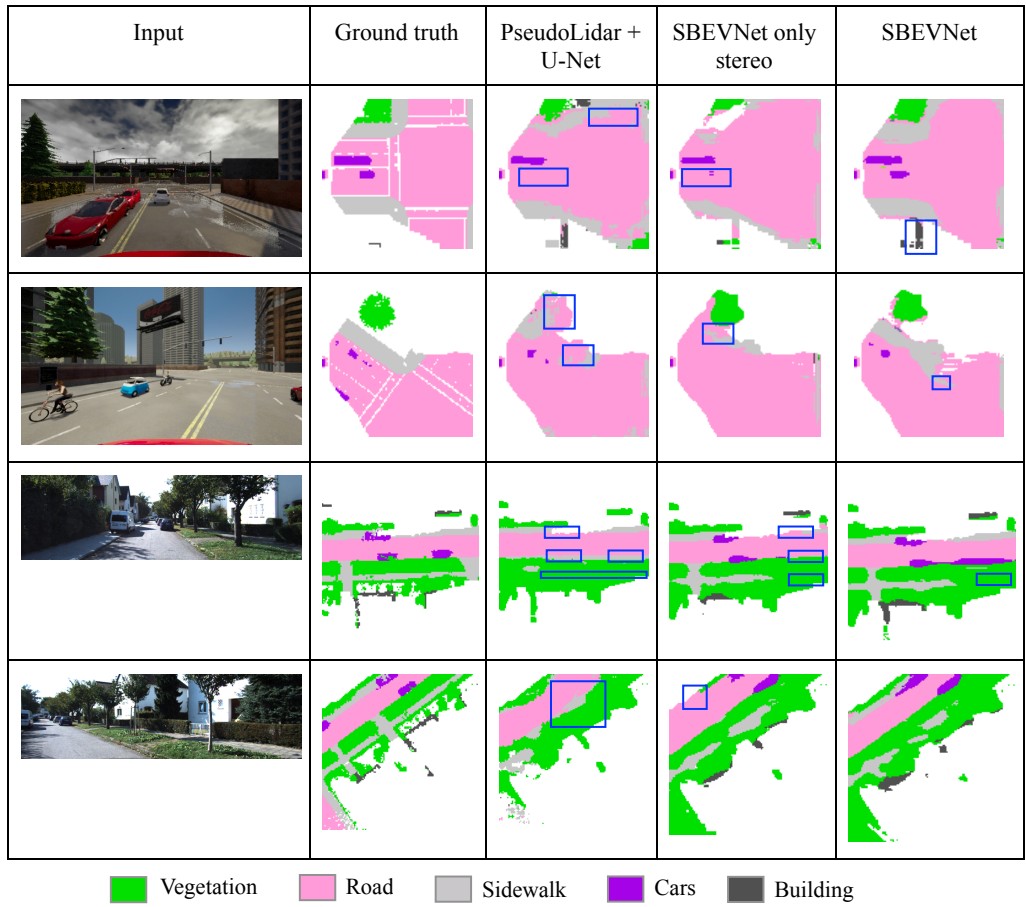

Figure 2: Qualitative results on the test set of the CARLA and the KITTI dataset. The major mistakes in the predictions are annotated by a blue rectangle.

We first compare our method with Pseudo-lidar (Wang et al., 2019a), where we use PSMNet to generate a 3D point cloud. In **Pseudo-LiDAR + segmentation** we project the semantic segmentation of the front view adn project it to the bird's eye view. In **Pseudo-LiDAR + BEV U-Net**, the RGB 3D point projected in the BEV space is fed as an input to the U-Net segmentation network. In **IPM + BEV U-Net** inverse perspective mapping is applied to the input image, which is fed as an input to the

| Method | mIoU | Road | Sidewalk | Cars | Building | Vegetation |
|---|---|---|---|---|---|---|
| Pseudo-LiDAR + segmentation | 18.69 | 35.51 | 14.56 | 13.50 | 12.64 | 17.26 |
| Pseudo-LiDAR + BEV U-Net | 28.97 | 61.83 | 21.36 | 5.55 | 12.74 | 43.38 |
| IPM + BEV U-Net | 34.93 | 68.40 | 26.65 | 28.49 | 4.53 | 46.57 |
| MonoLayout | 25.19 | 64.36 | 20.53 | 2.43 | 2.59 | 36.05 |
| MonoLayout + depth | 21.48 | 55.80 | 16.19 | 1.91 | 3.03 | 30.46 |
| MonoOccupancy + depth | 29.16 | 70.52 | 22.17 | 7.11 | 5.25 | 40.77 |
| *SBEVNet* only stereo | 50.01 | 78.41 | 40.16 | 41.96 | 30.45 | 59.05 |
| *SBEVNet* stereo + RGB IPM | 49.56 | 78.37 | 39.83 | 42.47 | 28.34 | 58.80 |
| *SBEVNet* stereo + features IPM | 50.60 | 80.16 | 41.08 | **43.64** | 29.19 | 58.92 |
| *SBEVNet*-CMD | 50.73 | **80.59** | 41.67 | 43.16 | 29.13 | 59.37 |
| ***SBEVNet*** | **51.36** | 80.23 | **41.86** | 42.81 | **31.35** | **59.43** |
| ***SBEVNet* Ensemble** | **53.85** | **82.22** | **45.70** | **44.97** | **34.54** | **61.83** |

Table 2: Quantitative results of semantic layout estimation on the KITTI dataset.

U-Net segmentation network. In **MonoLayout (Mani et al., 2020) + depth** and **MonoOccupancy (Lu et al., 2019) + depth**, we concatenate the input RGB image with the depth to train MonoLayout and MonoOccupancy model. Detailed description of the baselines can be found in the appendix.

We also evaluate some variations of our model to perform ablation studies. In *SBEVNet* **only stereo** we exclude the IPM features and only use features derived from the feature volume. To gauge the importance of IPM on RGB images and features, we also try applying IPM only on RGB images (*SBEVNet* **stereo + RGB IPM**) and IPM only on the features of the input image (*SBEVNet* **stereo + features IPM**). We also evaluate the cross modal distillation model *SBEVNet*-**CMD**. Finally, we evaluate our complete model (*SBEVNet* ) where we use stereo features and IPM on both RGB image and its features. We also evaluate *SBEVNet* **Ensemble** where we take an ensemble of *SBEVNet* with the same architecture but different initialization seeds.

## 4.4 EXPERIMENTAL RESULTS

We report the IoU scores of all the methods on the CARLA and KITTI (Geiger et al., 2013) dataset in Table 1 and Table 2 respectively. As we can see from the tables, *SBEVNet* achieves superior performance on both the datasets. We also observe the increase in performance if we use both stereo information and inverse perspective mapping. IPM yields a greater increase in performance in the CARLA (Dosovitskiy et al., 2017) dataset because the ground is perfectly flat. If we use only RGB IPM along with stereo, the results are slightly worse on the KITTI dataset because the ground is not perfectly planar. We see that degradation does not persist if we also use IPM on the image features. For the KITTI dataset, we see a sharp improvement over pseudo-LiDAR approaches because of inaccurate depth estimation. On the other hand, our model does not depend on explicit depth data/model. The results of MonoLayout (Mani et al., 2020) and MonoOccupancy (Lu et al., 2019) are inferior due to lack of any camera geometry priors in the network. We also show the qualitative results on the test set of CARLA (Dosovitskiy et al., 2017) and KITTI (Geiger et al., 2013) dataset in Figure 2. We see that in certain regions *SBEVNet* gives outputs closer to the ground truth. More details and analysis of the results can be found in the appendix.

## 5 CONCLUSION

In this paper we proposed *SBEVNet*, an end-to-end network to estimate the bird's eye view layout using a pair of stereo images. We observe improvement in the IoU scores compared with approaches that are not end-to-end or do not use geometry. We also showed that combining inverse perspective mapping with the projected disparity feature volume gives better performance. We also show that, using cross modal distillation to transfer knowledge from IPM features to the stereo features gives us an improvement in results.

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

## A  APPENDIX

### A.1  DATASETS

**CARLA dataset:** We use the CARLA (Dosovitskiy et al., 2017) simulator to generate a synthetic dataset. A pair of stereo cameras are placed on a car moving along a set of waypoints. We also randomly change the weather/lighting conditions. We use the point cloud of the simulator's city model to generate the ground truth semantic map. To ensure that we evaluate the generalizability of the models, the training and testing are done on entirely different city models in CARLA. Town01, Town02, Town03, and Town04 are used for training, and Town05 is used for testing. The training set contains 4,000 pairs of stereo images and the testing set contains 926 pairs of stereo images. The classes we use for the semantic map are road, vegetation, car, sidewalk, and building. The bounds of the layout with respect to the camera are -19 to 19 meters in $x$ direction and 1 to 39 meters in the $y$ direction.

**KITTI dataset:** We also evaluate *SBEVNet* on the publically available KITTI (Geiger et al., 2013) dataset. Similar to Mani et al. (2020), we use the KITTI odometery subset and use the SemanticKITTI Behley et al. (2019) dataset for labeled ground truth point clouds. We use the same training/testing split as used by Mani et al. (2020), where separate sequences are used for training

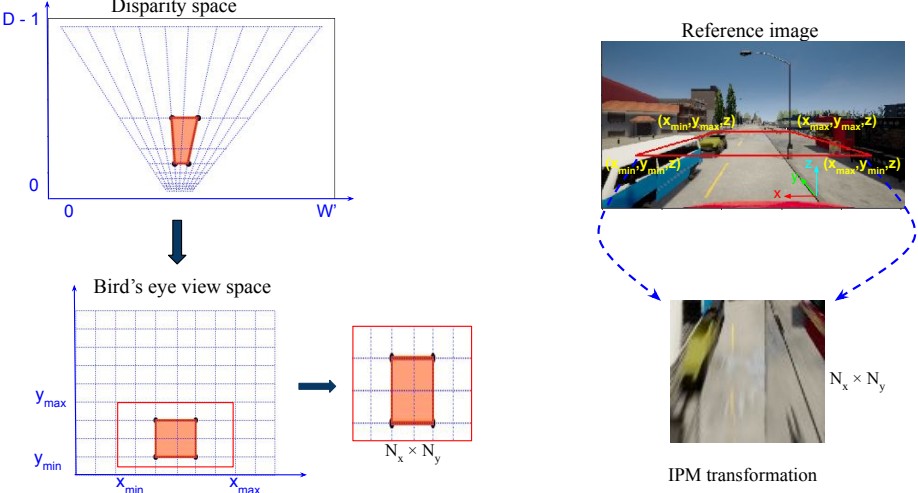

Figure 3: Illustration of mapping the disparity space to bird's eye view space and inverse perspective mapping. (a) The operation maps different disparities and $x$ to the BEV space in order to match the ground truth. We also show an example layout warped to the disparity space. (b) The inverse perspective mapping operation maps pixels of the reference image to the BEV space in order to match the ground truth. The same mapping can be applied to the image features as well.

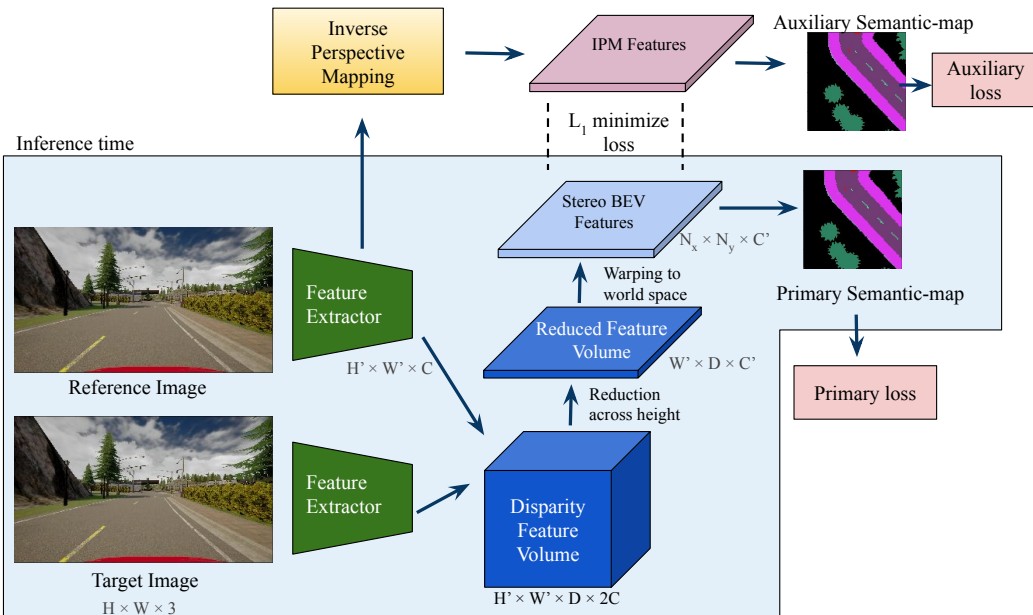

Figure 4: *SBEVNet*-CMD overview. We first extract the image features given the target and reference image. Using the pair of features, we create a stereo BEV representation. During training time, we apply inverse perspective mapping (IPM) on the image features which is used to predict the BEV layout separately. We minimize the L1 distance between first K channels of both the BEV feature maps. During inference time, we only use the stereo BEV representation to estimate the semantic map.

and testing. The training set contains 3,278 stereo image pairs and the testing set contains 1,371 stereo image pairs. The classes we use for the semantic map are road, vegetation, car, sidewalk, and

building. The bounds of the layout with respect to the camera are -19 to 19 meters in $x$ direction and 5 to 43 meters in the $y$ direction.

## A.2 COMPARED METHODS

1. **Pseudo-LiDAR (Wang et al., 2019a) + segmentation**: Uses Pseudo-lidar with PSMNet to generate a 3D point cloud from the input stereo images which is used to project the semantic segmentation of the front view to the bird's eye view. The PSMNet is trained separately on the respective datasets for better performance.

2. **Pseudo-LiDAR (Wang et al., 2019a) + BEV U-Net**: The RGB 3D point projected in the BEV aligned with the ground truth layout is used to train a U-Net segmentation network.

3. **IPM + BEV U-Net**: Inverse perspective mapping is applied to the input image to project it to the BEV space which is used to train a U-Net segmentation network.

4. **MonoLayout (Mani et al., 2020)**: This baseline uses MonoLayout to generate BEV semantic map from a single image. Rather than using OpenStreetMap data for adversarial training, we used random samples from the training set itself.

5. **MonoLayout (Mani et al., 2020) + depth**: The input RGB image concatenated with the depth is used as an input to the MonoLayout Model.

6. **MonoOccupancy (Lu et al., 2019) + depth**: The input RGB image concatenated with the depth is used as an input to the MonoOccupancy Model.

## A.3 IMPLEMENTATION DETAILS

We implemented *SBEVNet* using Pytorch. We use Adam optimizer with the initial learning rate of 0.001 and betas (0.9, 0.999) for training. We use a batch-size of 3 on a Titan X Pascal GPU. We use the same base network which is used in the basic model of PSMNet. The input image size for the CARLA dataset is 512×288 and the input image size for the KITTI dataset is 640×256. We report the average scores according to 8 runs to account for the stochasticity due to random initialization and other non-deterministic operations in the network.

## A.4 EXPERIMENTAL RESULTS

### A.4.1 ABLATION STUDY

**IPM on RGB image** For the CARLA dataset, we observe an increase of 3.67 in the mIoU score, on concatenating IPM RGB with the stereo features. We observe an increase in IoU scores for all the classes, with the biggest increase of 8.12 in the cars class. For the KITTI dataset, there is a small decrease of 0.45 in the mIoU score. This is because, the ground is not perfectly planar, hence the IPM RGB images do not exactly align with the ground truth layout.

**IPM on image features** If we apply IPM on the features of the input image and concatenate it with the features from the stereo branch, we see an improvement in both the datasets. The improvements in mIoU scores are 6.19 and 0.59 for the CARLA and KITTI dataset. The improvement is higher compared to the RGB IPM because image features contain higher level information which is transformed to the BEV space.

**IPM on both RGB image and image features** We see the greatest improvement if we apply IPM on both the RBG image and the features of the RGB image. The improvements in mIoU scores are 8.26 and 1.35 for the CARLA and KITTI dataset respectively. This is because the model is able to exploit the different information present in $R_{\text{IPM\_feat}}$ and $R_{\text{stereo}}$.

**Cross modal distillation** The performance of *SBEVNet*-CDM is in between of stereo only *SBEVNet* and full *SBEVNet*. We see an improvement of 4.00 and 0.72 in the mIoU scores on the CARLA and KITTI dataset, if we train the stereo model using cross modal distillation via IPM features. During inference, the architecture of *SBEVNet*-CMD is the same as the stereo only *SBEVNet*. This shows that CMD is able to transfer most of the IPM knowledge to the stereo branch.

**Minimizing distance between first $K$ features** We also evaluate the approach, where we try minimizing the L1 distance between all the channels of the IPM features and stereo features. We observe

mIoU scores of 32.27 and 50.03 for the CARLA and KITTI dataset respectively. This is worse than the mIoU scores achieved by minimizing the distance between first $K$ channels. This is because, if we enforce all stereo branch channels to be the same as IPM branch channel, the stereo branch is unable to learn information that is not present in the IPM features.

## A.5 ADDITIONAL ANALYSIS

### A.5.1 3D FEATURE VOLUME ANALYSIS

One claim of our approach is that our model learns 3D information without any explicit depth/disparity supervision. To validate this claim, we use the learned 3D feature volume to perform disparity estimation. We freeze all the weights and add a small 3D convolution layer to perform disparity regression on the learned feature volume. We also observe that the feature volume which is trained with cross modal distillation via IPM performs better at the task of disparity estimation. For the CARLA dataset, we find that the *SBEVNet* only stereo model has a 3-pixel error of **7.92** and with cross model distillation the 3-pixel error goes down to **6.84**.

### A.5.2 DISTANCE FROM CAMERA

We wish to quantify how our system performs as we move away from the camera. Hence, we plot the IoU scores for the pixels in the BEV layout which are more than a given distance from the camera (Figure 6) and for the pixels which are less than a given distance from the camera (Figure 5). For both the KITTI and the CARLA dataset, we observe that there is a drop in performance as the distance from the camera increases. We also observe that *SBEVNet* outperforms the stereo only *SBEVNet* at all distances from the camera.

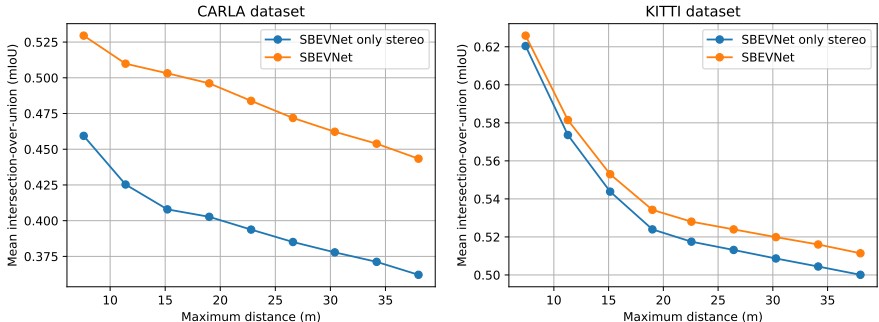

Figure 5: Performance as a function of maximum distance from the camera. We consider the pixels in the BEV layout which are atmost a certain distance away from the camera.

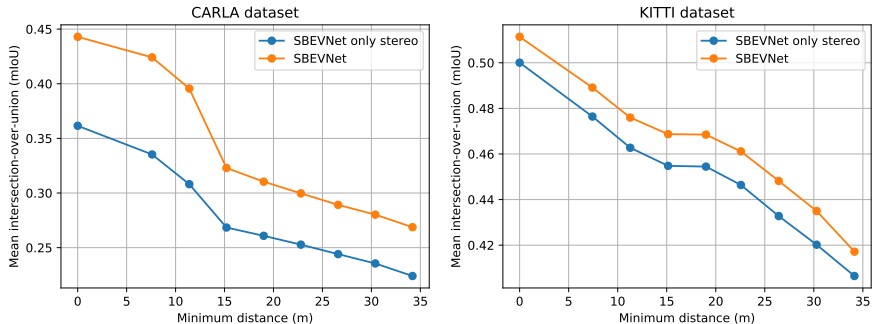

Figure 6: Performance as a function of minimum distance from the camera. We consider the pixels in the BEV layout which are atleast a certain distance away from the camera.

### A.5.3 AMOUNT OF TRAINING DATA

We also quantify how the performance of *SBEVNet* changes with the number of data-points in the training set, while keeping the test set same. On the CARLA dataset, with just 10% of the training data, we get mIoU score of 26.10 compared to the mIoU score of 44.36 with all the training data. For both the datasets, performance starts to saturate when we use 100% of the training data. This shows that 3,000-4,000 training data points are sufficient for getting the optimal performance from *SBEVNet*.

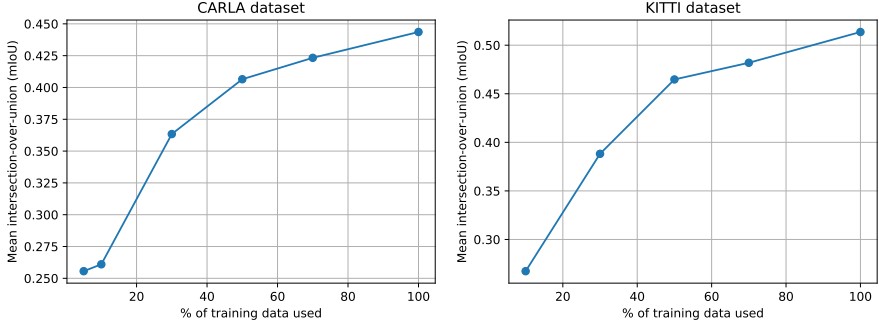

Figure 7: Performance of the system as a function of amount of training data used.

### A.5.4 PERFORMANCE EVOLUTION DURING TRAINING

We also perform a qualitative analysis (Figure 8) of how the performance of the model changes during training. We observe, during the initial stages of training, the model learns to identify very course grained attributes such as the direction of the road. However, there is ambiguity in detailed attributes such as size and exact position. During the later stages of training, the model learns to identify the smaller objects and exact positioning in the BEV space.

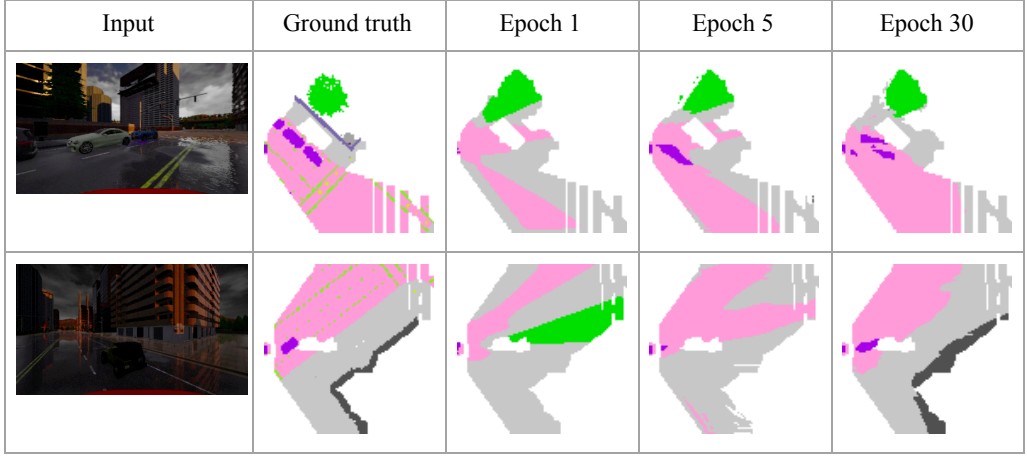

Figure 8: Evolution of predicted BEV layouts for different epochs during training

### A.5.5 ENSEMBLE

We observe some variance in the performance of the models on training with different random seeds. For *SBEVNet* we observe a standard deviation of 2.16 and 2.46 in the mIoU scores for the KITTI and CARLA dataset respectively. Due to the diversity in outputs of the individual models (Zhou (2012)), we see an improvement in the performance, if we take an ensemble of individual models.

We observe an absolute improvement of 2.49 and 3.56 in the mIoU scores for the KITTI and CARLA dataset respectively.

### A.5.6 INFERENCE TIME

On NVIDIA Titan X GPU, with the batch size equal to 1, the inference time of stereo only *SBEVNet* for one input pair is 0.1307s on the average. For the full *SBEVNet* the inference time is 0.1449s on the average, which is slightly higher than the stereo only model. This inference speed is sufficient for majority of robotics applications. The majority of computation is done in processing the 3D feature volume with 3D convolutions.

### A.6 RESULTS ON KITTI OBJECT DATASET

We also compare our method with the published numbers on the KITTI Object dataset. We use the dataset and annotations provided by Mani et al. (2020). Here, there is only a single cars class. We compare our approach with published monocular approaches and 3D object detection approach on pseudo lidar with stereo input. AVOD + pseudo lidar is an object detection method which also uses the large sceneflow dataset for pre-training. More details of these baseline models can be found in Mani et al. (2020). Table 3 shows the numbers on the KITTI object split which are provided by Mani et al. (2020). We observe that our method achieves an improvement in the mIoU scores over all the other methods. For segmentation, pixel level mAP is not a good metric as is does not consider false negatives. We still report the pixel level mAP scores for reference.

| Method | mIoU | mAP (pixel level) |
|---|---|---|
| ENet + Pseudo lidar input(Monodepth2) PointRCNN | 0.24 | 0.37 |
| PointRCNN + Pseudo lidar input(Monodepth2) | 0.26 | 0.43 |
| MonoLayout | 0.26 | 0.41 |
| AVOD + Pseudo lidar input(PSMNet) (Stereo) | 0.43 | **0.59** |
| **Our (Stereo)** | **0.46** | **0.59** |

Table 3: Quantitative results of BEV car segmentation on the KITTI object dataset. All the results except *SBEVNet* are excerpted from Mani et al. (2020)

