# OpenReview forum: "SBEVNet: End-to-End Deep Stereo Layout Estimation"
_ICLR.cc/2021/Conference — Reject_

### Official Review · AnonReviewer4 · 2020-10-28
**SBEVNet Review**

**Rating:** 5
**Confidence:** 4

**Review:**

The paper proposes an end-to-end network for layout estimation from stereo images. The approach is built off previous stereo matching networks, which built and process a 3D disparity volume. The stereo estimate is used to project image features into a birds-eye-view representation which is processed using a U-net which predicts a semantic scene layout. The approach is evaluated on the KITTI and Carla generated datasets.

Strengths:
* This is the first work to attempt semantic layout estimation from stereo images
* The approach is geometrically grounded, and can properly leverage stereo information to improve layout estimation
* The approach performs well on the two datasets evaluated. Since this paper is focused on a new problem, there are not existing works to directly compare to. However, the paper provides reasonable baselines by modifying existing networks for this task
* Avoids the need for an intermediate representation (i.e. point cloud) by directly mapping features from the disparity volume into birds-eye-view coordinates
* Plots in the appendix are interesting

Weaknesses:
* While the task itself is new, closely related forms of the problem have been studied. For example 3D object detection from monocular/stereo, and monocular layout estimation. It would have been helpful to see results on the closely related task of 3D object detection to better compare against prior works.
*  The IPM module appears to be very sensitive to the accuracy of the ground plane. In the synthetic CARLA dataset, where a ground plane can be accurately computed, there seems to be a large advantage of using the IPM module. On real-world data like KITTI, the use of the IPM module gives very limited improvement in performance of the stereo-only baseline.
* The task is closely related to 3D object detection which has been using similar components. The core components of the approach have been used in various forms in prior work. The paper (Orthographic feature transform for monocular 3d object detection, Roddick 2019) uses a very similar method to project image features into a birds-eye-view representation.

---

> ### Author Response · Authors · 2020-11-16
> **Response to reviewer #4**
>
> Thank you for your comments and the review of the paper.
>
> We have compared our model on the KITTI Object dataset with some prior works for 3D object detection, which is available in the appendix.
>
> There is a key difference in our IPM module and the Orthographic Feature Transform proposed in Roddick 2019. Even though both the methods project the input image features to the birds eye view space, OFT does not use any ground information. Because of that, there is a many-to-one mapping from the image space to the bird's eye view space. IPM uses ground information to compute a homography which transforms image features to BEV in a one-to-one fashion.

---

### Official Review · AnonReviewer2 · 2020-10-28
**Interesting problem but the paper can be improved**

**Rating:** 6
**Confidence:** 4

**Review:**

Interesting problem but the paper can be improved

This work aims to directly estimate the world layout in front of the vehicle from a pair of stereo cameras. It is based on cost volume but it does not explicitly predict the depth values of each pixel. Instead, it warps the cost volume features to the bird eye view (BEV) and do semantic segmentation from BEV using U-Net.

I think the problem is interesting and I believe it has never been mentioned and/or addressed before. Moreover I think the motivation is also valid, as the BEV semantic segmentation from camera sensors can be one important perception input for navigation and planning.

I like the idea of skipping the explicit 3D reconstruction and directly shoot for the final goal; I believe we usually get better performance when we directly minimize the loss we want to minimize. Moreover, it could potentially introduce some inspiration to other works, e.g., the direct extension - 3D (point/volume) semantic segmentation.

Though I like this work, I also has several concerns:
1. Is IPM feature really important? I only see it is effective on the synthetic dataset CARLA but not on KITTI. What is the possible reason? My guess is that the ground estimation is very bad for the real-world data. I am also curious what is the performance if only IPM feature is used.
2. In introduction, this paper claims estimating accurate depth is not sufficient due to occlusion. However, I don't see how this work could handle occlusion. Instead the occluded part is masked out during training. Please explain this statement.
3. What is the range of the layout estimation? From CARLA, it is 39m, and from Figure 5 and Figure 6, it is 35m. If it is the case, the short range of the estimation makes it hard to act as a major component in the perception system; the best use case is for short range detection and system redundancy. But actually I can imagine that it would not get very good result in long range, as there is always a trade-off between baseline (for accuracy) and camera overlap (for coverage) in stereo estimation.
4. What is the image resolution for the inference time test? It seems quite slow if the resolution is 512x288 (for CARLA) or 640x256 (for KITTI).
5. For the experiment, I think it is better to report mean $\pm$ std with multiple trainings, as there are training noise.

Other suggestions and clarifications
1. When IPM is first introduced in Page 2, it is better to explain it in a short sentence. The current version is not clear and there are typos.
2. I believe this work is based on binocular stereo pairs (correct me if I am wrong), so please explicitly say that in the paper. Also, using left/right image instead of reference/target image is less misleading.
3. For disparity feature volume, it is better to use the prevalent name - cost volume. It is called cost volume in the introduction but later called disparity feature volume, I think it is better to be consistent.
4. It is unclear how IPM feature are obtained: from pre-determined parameters or ground estimation? I think pre-determined parameters will not work very well because ground is not always a perfect plane.
5. It is unclear what is the ensemble method used here. If it just takes the best of several models, I will not be convinced.

After rebuttal:
I still think this work has a interesting task setup, though it indeed has many faults (after reading the responses and other reviewers):
1. It seems that IPM is not really useful in practice.
2. It is also not sufficient to large occlusion, and thus there is no explanation for its advatange over `estimating accurate depth`
3. Range is short and latency is high
4. After reading reviewer1's comments, I think it could use the same experimental setting as the existing methods for a fair comparison. The other methods might be not properly trained with the new setting.
5. It is still not clear how to emsemble several models (with different trained weights) in this work.
Thus I am changin my rating to 6, and I will not fight for this work.

---

> ### Author Response · Authors · 2020-11-16
> **Response to reviewer #2**
>
> Thank you for your comments and the review of the paper.
>
> - There is very less improvement by using IPM on the KITTI dataset because the ground is not perfectly flat in real world.
>
> - We only mask out strong occlusions things which cannot be "guessed" by the model without looking. Whereas weak occlusions ( where the occluded area is small ) are not masked out. Weakly occluded areas ( eg things behind a thin tree)  are easy to guess by the model.
>
> - For the CARLA dataset, the bounds of the layout with respect to the camera are -19 to 19 meters in x direction and 1 to 39 meters in the y direction. For the KITTI dataset, the bounds of the layout with respect to the camera are -19 to 19 meters in x direction and 5 to 43 meters in the y direction.
>
> - The image resolution at inference is 512x288 and 640x256 and the inference time is about s 0.1449s on a TitalX GPU. Most of the inference time is used in feature volume generation. Our approach should work with newer approaches such as high-res-stereo for faster feature volume generation.
>
> - Thanks, we will change the introduction in Page 2 and we will make it  more clear that this work uses binocular stereo.
>
> - Using the term "cost volume" for our model could be misleading because we are not computing the matching costs. We only used the term "cost volume" to describe other works for disparity estimation where matching costs at each disparity is computed.
>
> - For the CARLA dataset, the ground estimation is done using pre-determined parameters. For the KITTI dataset, the plane is computed using the depth values of all the "road" pixels. The depth values are computed using standard stereo matching algorithm and the "road" pixels are optioned using a standard pre-trained semantic segmentation network.
>
> - The ensemble method, takes an ensemble of the same model initialized by different seeds.

---

### Official Review · AnonReviewer3 · 2020-10-29
**This paper proposed to estimate the semantic layout in the bird eye's view from stereo images. The main novelties lie in how to organize and exploit the information from the stereo images. The proposed framework builds upon inverse perspective mapping, projected stereo feature volume. The performance was evaluated on the KITTI and CARLA datasets.**

**Rating:** 5
**Confidence:** 5

**Review:**

The paper proposed to estimate the semantic layout in the bird eye's view from a pair of stereo images. The main novelty/contribution lies in how to organize and exploit the information from the stereo images. The proposed framework builds upon inverse perspective mapping, and projected stereo feature volume. The performance was evaluated on the KITTI and CARLA datasets. Given a pair of stereo images, there are various options to exploit the image information, where this paper provides a framework by exploiting the stereo information in the bird eye's view.

- A principled question is what is the real superiority of estimating the layout in the bird eye's view. From the application' view, the semantic estimation from the camera'view already provide much information which the stereo images could further improve the performance. From the applications's perspective, I would like to see discussions and experiments in showing the superiority in using the bird eye's representation.

- In the ablation studies, the paper already provide different variants of the network architecture in exploiting the stereo image information. I believe there are multi-task learning based framework form this task, where the semantic layout estimation and stereo estimation are jointly estimated and optimized. Whether that pipeline will provide extra benefit?

- In section 3.4.4, the paper claimed that "We pass the concatenated stereo BEV feature map and IPM BEV feature map to a U-Net (Ronneberger et al., 2015) network to generate the semantic map C". However, the loss evaluation applies to the IPM features and Stereo features separately, namely, $\mathcal{C}_i^{IPM}$ and $\mathcal{C}_i^{Stereo}$. If two estimations are made as the network output, which one will be used for performance evaluation? The other following question is : If two separate estimations are made as the network outputs and compared with the ground truth for loss evaluation, whether a consistency loss between these two estimations will further constrain the network learning?

- The paper conducted experiments on the KITTI and CARLA dataset. It is well understood that the CityScape dataset has been widely in evaluating semantic segmentation where the stereo images are available. I would to see more evaluation on these real-image dataset rather than synthetic dataset such as the CARLA dataset.

- The paper title and abstract should highlight "semantic" and "bird eye's view" as the paper proposed to learn the semantic layout in the bird eye's view. The current title did reflect these properties.

All in all, taking all the above comments into consideration, I would like to hear from the authors' response, which could lead to updated rating in either directions.

---

> ### Author Response · Authors · 2020-11-16
> **Response to reviewer #3**
>
> Thank you for your comments and the review of the paper.
>
> - For the application of navigation and path planning, bird's eye layout is very useful because we want to plan the exact x,y coordinates of the path of the robot to reach a given goal.
>
> - In our initial experiments we did not observe a big difference in performance when stereo disparity estimation is used as auxiliary task in the multi-task learning setting.
>
> - As mentioned in the paper, CiIPM and CiStereo  losses are applied only for the SBEVNet-CMD model. There the IPM features are only used as an auxiliary task during training and only stereo features are used during inference.  For the SBEVNet-CMD model, a consistency loss is applied only of the first K features of the representations which does not fully constrain the network training.  However the main SBEVNet model uses the loss mentioned in the equation 6, which uses a concatenated stereo and BEV features.
>
> - Unfortunately, we could not run evaluate our model on the CityScape dataset due to lack of dense semantically labeled point cloud ground truth or bird's eye view layout ground truth.

---

### Official Review · AnonReviewer1 · 2020-10-30
**Concerns about experiment setup/design**

**Rating:** 5
**Confidence:** 5

**Review:**


## Contributions

This paper presents SBEVNet, a neural network architecture to estimate the bird's-eye view (BEV) layout of an urban driving scene. Given an image captured by a stereo camera, SBEVNet performs an inverse perspective mapping (IPM) to obtain an initial feature volume, which is further processed to generate the BEV layout. The system is trained end-to-end in a supervised learning setup.

## Strengths

**S1** The problem considered here is very relevant to perception groups in the autonomous driving community. This area has only recently seen work crop up. Approaches like MonoLayout [A], MonoOccupancy [B], and PseudoLidar [C] are closely related to this submission.

**S2** The paper is easy to follow, and provides a majority of the details needed to understand and assess the approach.

**S3** The authors also seem to provide code (and promise a public release), which might help ensure reproducibility.

## Weaknesses

I see a few major and a number of other minor concerns that impact my perception of this paper. I'm hoping the discussion period helps address some of these, and I'm open to revising my score in light of evidence contrary to the following claims.

It appears that this paper uses MonoLayout [A], MonoOccupancy [B], and PseudoLidar [C] as primary baselines. Much of my review stems from my understanding of [A, B, C] (and my 'surprise' at a few contradictory trends observed in this paper.)

**Problem setup** It is unclear from reading the paper and supplementary material if the problem setup is infact "amodal" layout estimation (i.e., if scene points outside of the camera view are predicted in the BEV layout). Approaches like (Schulter et al., 2016) and (Mani et al., 2020) operate in this "amodal" setup, while others such as PseudoLidar [C] and (Lu et al., 2019) only predict points that are visible in the input image. Does this approach, for instance, hallucianate hidden intersections and roads? (It seems not, since a visibility mask is explicitly employed in the loss function -- cf. Fig. 1 and Eq. 12, 13).

**MonoLayout baseline** The primary baseline considered in this paper is "MonoLayout" (Mani et al., 2020). Upon examining the MonoLayout [A] paper, I find a surprising and troubling trend. This paper reports very poor performances of MonoLayout on the KITTI dataset (the original MonoLayout paper reports mIoU for the "car" class to be around 26.08, while the current submission reports 2.43 -- cf. Table 2). I've noted that MonoLayout makes its code and models publicly available (its publicly available pretrained models claim an mIoU of 30.18 for the "car" class), as highlighted on their GitHub page. Also, other baselines like "MonoOccupancy" have surprisingly low scores in this paper (an order of magnitude), compared to scores reported in the MonoLayout paper. I wonder if there is something different in the experiment and/or training protocols employed in the current work, as opposed to those in the MonoOccupancy and MonoLayout papers? For example, the MonoOccupancy baseline as reported in the MonoLayout paper achieves an mIoU of about 24.16 (for the car class) (MonoLayout paper - Table 1), while the same baseline has a dismal performance (mIoU of 7.11 for car class) in Table 2 of the current manuscript.

The fact that this performance gap is not explained in the paper makes it hard to analyze the merits of the proposed approach. Save for a single sentence "The results of MonoLayout ... and MonoOccupancy ... are inferior due to lack of any camera geometry priors in the network", I've not found any other discussion of this performance gap/discrepancy.

I also find it a tad weird (and unexplained) that the performance of various baselines do not seem to follow a set pattern/trend across the CARLA and KITTI datasets. In the MonoLayout paper, I notice that changing the dataset from KITTI to Argoverse does change absolute mIoU scores a bit, but preserves the ranking of various baselines (i.e., MonoLayout > OFT > MonoOccupancy on both KITTI and Argoverse). In the current submission, the trends seem to be changing across the two datasets (cf. Tables 1, 2).

Yet another set of baselines that seem to underperform here are the PseudoLidar variants. In the MonoLayout paper (cf. supplementary material, Table 5), Pseudolidar is evaluated on the KITTI dataset, and the reported mIoU for vehicles is 59, whereas in this paper the best performance on this class achieved by a pseudolidar model is 45.64. Further, the MonoLayout paper's version of the (stereo) Pseudolidar baseline seems to perform quite competetively (mIoU 59.0) to SBEVNet Ensemble (mIoU 60.17 for "car", cf. Table 2). This seems to indicate that well-tuned baselines could perhaps achieve better performance?

In Appendix A.2, the authors seem to indicate that they used a very different process to train MonoLayout (i.e., using random images from the train set as opposed to using OpenStreetMap and/or adversarial training). I suspect this might have resulted in a performance gap?

I feel that OFT [D] could be cited and used as a baseline, particularly to measure layout estimation accuracy for the "car" class.

**Qualitative results** Unfortunately, there seems to be a dearth of qualitative result figures to get a better sense of the approach. In particular MonoLayout and MonoOccupancy seem to obtain crisp reconstructions of cars (cf. MonoLayout paper), while in Figure 2., cars are splayed throughout the image in the SBEVNet results. This is also surprising; in my opinion, these results do not adequately substantiate the impressive reported mIoU.

**Missing mAP metric** Other papers such as MonoLayout and OFT seem to report the mAP (mean average precision) metric in addition to the mIoU metric, because mAP often turns out to be a more accurate estimate of prediction performance (due to integrating over various recall values). In practice, this leads to less-than-perfect predictions being scored well (and this could explain the splayed-out results in Fig. 2 scoring a high mIoU). Evaluating mAP would be a stricter criteria, and will allow an additional point of comparison with prior art.


## Minor remarks

The following remarks have had no impact on my assessment of the paper, and as such I don't expect the authors to respond to these.

Concurrent approaches such as [F] can be cited and discussed.

The paper could be structured better. For instance, input image sizes and baselines could be moved over to the main paper, rather than being listed in the appendix.

## References

[A] Mani, Kaustubh, et al. "MonoLayout: Amodal scene layout from a single image." The IEEE Winter Conference on Applications of Computer Vision. 2020.

[B] Lu, Chenyang, Marinus Jacobus Gerardus van de Molengraft, and Gijs Dubbelman. "Monocular semantic occupancy grid mapping with convolutional variational encoder–decoder networks." IEEE Robotics and Automation Letters 4.2 (2019): 445-452.

[C] Wang, Yan, et al. "Pseudo-lidar from visual depth estimation: Bridging the gap in 3d object detection for autonomous driving." Proceedings of the IEEE Conference on Computer Vision and Pattern Recognition. 2019.

[D] Roddick, Thomas, Alex Kendall, and Roberto Cipolla. "Orthographic feature transform for monocular 3d object detection." arXiv preprint arXiv:1811.08188 (2018).

[E] A Parametric Top-View Representation of Complex Road Scenes. CVPR 2019.

[F] Lift, Splat, Shoot: Encoding Images From Arbitrary Camera Rigs by Implicitly Unprojecting to 3D. ECCV 2020.

---

> ### Author Response · Authors · 2020-11-16
> **Response to reviewer #1**
>
> Thank you for your comments and the review of the paper.
>
> - The areas which are strongly occluded are masked out and are not used for evaluation. Whereas, weak occlusions ( where the occluded area is small ) are not masked out. Weakly occluded areas ( eg things behind a thin tree)  are easy to guess by the model.  For a fair comparison, in our main tables, we evaluate all the baselines using the same visibility mask.
>
> - We observed a huge drop in performance if we increase the number of semantic classes in the official code of the MonoLayout dataset. Another difference is the subset of the KITTI datasets used for experiments. Monolayout reports their numbers for "car" class on the KITTI object dataset, whereas our "car" class is also for the KITTI  odometry split. We chose the KITTI odometry spit because we have annotations of other classs other than "car". But for a fair comparison, we also train and compare our model ( mentioned on the Table 3 of the appendix)  on the KITTI Object split where we follow the exact same protocol as Monolayout. The difference is because of total number of classes is different, the split is diffrent. Monolayout also has a large number of hyperparamaters ( for eg class weight )  which are hard to tune for datasets which have more classes.
>
> - The reason for not having the same trend in the baselines is because of the differences in the KITTI and the CARLA dataset. The CARLA dataset is synthetically generated dataset where the ground is perfectly flat. On the other hand, the KITTI dataset is a real world dataset with non perfect ground and non perfect BEV annotations.
>
> - We have compared the numbers for PseudoLidar published by Monolayout in the Table 3 of the appendix. The difference in results is because they are diffrent subsets of the KITTI dataset and with diffrent number of classes in the BEV. We observed a superior performance in out model on the KITTI Object subset compared to the published numbers.
>
> - mAP metric is widely used for the task of object detection ( models like OFT ) where predicted bounding boxes are evaluated over various IoU values. On the other hand, we  predict a pixel-wise semantic semantic segmentation mask in the birds eye view. We had not used mAP because most of the works doing semantic segmentation do not use mAP as an evaluation metric.
>
> The function to compute the mAP in the official code of Monolayout, only computes just the pixelwise precision for each image and take an average ( check out [utils.py](http://utils.py) of their official code) . They do not integrate over various recall value. This is not a good way because a good score can be achieved reducing false positives by lowering the threshold.

---

### Decision · Program_Chairs · 2021-01-07
**Final Decision**

**Decision:**

Reject

**Comment:**

This paper addresses the problem of estimating a “birds-eyed-view” overhead semantic layout estimate of a scene given an input pair of stereo images of the scene. The authors present an end-to-end trainable deep network that fuses features derived from the stereo images and projects these features into an overhead coordinate frame which is passed through a U-Net style model to generate the final top view semantic segmentation map. The model is trained in a fully supervised manner. Experiments are performed on the CARLA and KITTI datasets.

While R2 was positive, they still had some concerns after reading the rebuttal and the other reviews. Specifically, they were not convinced about the value of the IPM module. This concern was also shared by R4, especially in light of the relationship to Roddick et al. BMVC 2019. R1 had concerns about the experiments, specifically the quantitative comparisons to MonoLayout. The authors addressed these comments, but it is still not clear if the differences can be attributed to the number of classes, how they are weighted, or the training split used? R3 had questions about the utility of BEV predictions in general. However, as stated by R2, there is a lot of value in approaching the problem in this way.

In conclusion, while there were some positive comments from the reviewers, there were also several significant concerns. With no reviewer willing to champion the paper, there is not enough support to justify accepting the paper in its current form.